# Myasthenia Gravis: An Acquired Interferonopathy?

**DOI:** 10.3390/cells11071218

**Published:** 2022-04-04

**Authors:** Cloé A. Payet, Axel You, Odessa-Maud Fayet, Nadine Dragin, Sonia Berrih-Aknin, Rozen Le Panse

**Affiliations:** Sorbonne University, INSERM, Institute of Myology, Center of Research in Myology, F-75013 Paris, France; cloe.payet@upmc.fr (C.A.P.); a.you@institut-myologie.org (A.Y.); odessa-maud@hotmail.fr (O.-M.F.); n.dragin@institut-myologie.org (N.D.); sonia.berrih-aknin@upmc.fr (S.B.-A.)

**Keywords:** interferon type I, autoimmunity, myasthenia gravis, thymus, thymoma, germinal center, sterile inflammation, pathogen infection, innate immunity, adaptive immunity

## Abstract

Myasthenia gravis (MG) is a rare autoimmune disease mediated by antibodies against components of the neuromuscular junction, particularly the acetylcholine receptor (AChR). The thymus plays a primary role in AChR-MG patients. In early-onset AChR-MG and thymoma-associated MG, an interferon type I (IFN-I) signature is clearly detected in the thymus. The origin of this chronic IFN-I expression in the thymus is not yet defined. IFN-I subtypes are normally produced in response to viral infection. However, genetic diseases called interferonopathies are associated with an aberrant chronic production of IFN-I defined as sterile inflammation. Some systemic autoimmune diseases also share common features with interferonopathies. This review aims to analyze the pathogenic role of IFN-I in these diseases as compared to AChR-MG in order to determine if AChR-MG could be an acquired interferonopathy.

## 1. Myasthenia Gravis

### 1.1. Generalities

Autoimmune myasthenia gravis (MG) is a rare neuromuscular disease due to an autoimmune attack against components of the neuromuscular junction. MG is a multifactorial disease resulting from a combination of genetic predisposition and environmental risk factors [1]. The disease can be classified according to the severity of the symptoms, age of onset, antigenic targets, and thymic-associated abnormalities. MG is characterized by fluctuating muscle weakness that can affect different muscles. Two major forms of the disease are described: an ocular form (15%) and a generalized form (85%). Symptoms of the ocular form are ptosis and/or diplopia. Generalized MG involves bulbar, limb, facial, and respiratory muscles. A respiratory crisis is the main life-threatening event that occurs in MG patients and leads to hospitalization [2].

Early-onset MG (EOMG) patients declare symptoms before the age of 45–50 years old and late-onset MG (LOMG) patients after 45–50 years old [3]. LOMG is often associated with thymoma. These subgroups display different features with female predominance for EOMG, an equal sex ratio for late-onset MG, and a male predominance for very late-onset MG (after the age of 65–70 years old) [3].

### 1.2. Autoantibody Targets Defining MG Subtypes

MG is due to autoantibodies targeting components of the neuromuscular junction on the postsynaptic membrane of striated skeletal muscles. Anti-acetylcholine receptor (AChR) antibodies are found in around 85–90% of generalized MG patients [4,5]. Though the symptoms are due to anti-AChR antibodies, the titer is not linked to the severity of the disease [6]. MG symptoms are also induced by autoantibodies against the muscle-specific kinase (MuSK) [7] and low-density lipoprotein receptor-related protein 4 (LRP4) [8]. MuSK-MG corresponds to 5% of the MG patients with a female predominance. They are more prone to severe and generalized muscle weakness. Unlike anti-AChR antibodies, anti-MuSK antibodies are predominantly of the IgG4 subtype and their titer is correlated with the severity of the disease [9]. LRP4-MG patients are even less frequent (2–5% of MG patients) and present with ocular or generalized MG, usually observed in young women [10]. Other autoantibodies are found in MG patients. They target agrin, titin, ryanodine receptors, rapsyn, cortactin, collagen Q, or collagen XII [11]. However, their pathogenic role is not defined yet. They usually co-exist with one of the three main autoantibodies involved in MG.

### 1.3. Implication of the Thymus in MG

The thymus is a primary lymphoid organ involved in T-cell differentiation and education. The thymus is divided into two main cellular zones: the cortex and the medulla, implicated in positive and negative thymocyte selections, respectively. It is composed of different stromal cells: thymic epithelial cells (TEC), fibroblasts, macrophages, dendritic cells (DC), and myoid cells [12]. Medullary TEC play an important role in tissue-specific antigen (TSA) presentation, essential for thymocyte selection and education to discriminate self- and non-self-antigens [13]. In this context, medullary TEC express the different subunits of AChR as TSA [14]. The thymus is the place where central tolerance and depletion of autoreactive T cells are established, to avoid autoimmunity. Of note, the myoid cells also express the different subunits of AChR and display a functional receptor [15]. In AChR MG patients, the thymus is considered as the effector organ and is very often abnormal as described below. No alteration is observed in the thymus of MuSK-MG patients [16] and the involvement of the thymus in LRP4-MG patients remains unclear so far [10,17].

The thymus does not present tissue abnormality in MuSK-MG patients [16] and so far, the involvement of the thymus in LRP4-MG patients remains unclear [10,17].

#### 1.3.1. Thymus of EOMG-AChR Patients

Thymic changes are clearly observed in EOMG patients but not in LOMG without thymoma, probably because the thymus significantly involutes with aging [18]. The EOMG thymus is characterized by B-cell infiltration, and 50–60% of them display follicular hyperplasia defined by the development of ectopic germinal centers (GC) in the thymic medulla [19,20]. GC are structures usually found in secondary lymphoid organs and are responsible for the proliferation, differentiation, and generation of memory B cells and plasma cells. B-cell infiltration plays a central role in MG development, and the number of GC is correlated with the autoantibody titer [21].

Myoid cells could play a role in the establishment of MG because they express a functional AChR. This cell population is decreased and colocalized with GC in MG patients [22]. Willcox et al. proposed that early anti-AChR antibodies present in the thymus target AChR in myoid cells, resulting in the activation of complement against those cells and the formation of GC [23]. Damage or destruction of those cells can lead to the release of AChR fragments and recapture by antigen-presenting cells, TEC or DC, potentially leading to sensitization against AChR.

In parallel with GC formation, neoangiogenesis processes have been observed in MG hyperplastic thymus. A high number of ectopic high endothelial vessels (HEV) is observed around GC and is correlated with the degree of hyperplasia [24]. Lymphatic vessels also develop in the MG thymus [25]. The increase in HEV and lymphatic vessels indicates strong cell trafficking between the thymus and periphery. This cellular trafficking is also controlled by the abnormal expression of certain chemokines in the MG thymus. Ectopic thymic HEV in MG express the chemokines CXCL12 and CCL17, responsible for the homing of CXCR4/ CXCR7 and CCR4 positive cells, respectively [24,26]. Thymic lymphatic vessels in MG thymus expressed CCL21, a CCR7 ligand, leading to the recruitment of circulating peripheral cells such as T cells and naïve B cells [25]. The overexpression of CXCL13 by TEC has been found in MG patients and could participate in the abnormal infiltration of B cells in the MG thymus [27]. Indeed, CXCL13 is a chemokine known to be implicated in the attraction of B cells and the formation of GC. Its receptor, CXCR5, is overexpressed on T follicular helper cells in MG thymus [28,29], which are important cells for antibody production in secondary lymphoid organs. MG TEC also overexpressed CXCL10, a chemokine that plays a role in the infiltration of T cells in inflamed sites. Its receptor, CXCR3, is also increased at the surface of T cells and leads to the recruitment of these cells in the thymus of MG patients [30].

These observations show that the hyperplastic thymus in MG displays numerous features normally observed in secondary lymphoid organs and due to its inflammatory status, it could be considered as a tertiary lymphoid organ. 

#### 1.3.2. Thymomas

Thymomas are thymic epithelial neoplasms and five main histological subtypes are defined according to the WHO classification: A = medullary thymoma, B1–2 = mainly or entirely cortical thymoma, AB = mix of medullary and cortical thymoma, and B3 = atypical thymoma [31]. Among thymoma patients, 90% develop an autoimmune disease, 30% of which are patients with MG associated with AChR-antibodies [32]. Thymoma-associated MG (MGT) is mostly of types B1 and B2. Inversely, among MG patients, only 10–20% have a thymoma. MGT affects males and females equally, usually individuals around 50 years old. MGT patients tend to have more severe symptoms as compared to EOMG patients [33]. B1 or B2 thymoma are characterized by the presence of immature thymocytes, and single positive CD4 or CD8 T can differentiate. However, the emergence of autoreactive T cells is probably due to altered negative selection process in a thymic disordered microenvironment, especially for B2 thymoma subtype with fewer medullar compartments. Thymoma is characterized by a deficit in the autoimmune regulator (AIRE) in neoplastic TEC [34]. AIRE controls the expression of numerous TSA in medullary TEC that is essential to establish central tolerance. This deficiency could impair negative selection of thymocytes. Indeed, thymoma patients can have autoantibodies involved in other autoimmune diseases, or against striational muscle autoantigens (such as titin, ryanodine) [34] or cytokines [35,36]. Likely related to this deficiency, patients have a lower number of regulatory T cells and a lower FOXP3 expression in the thymus that might affect peripheral tolerance [37]. Thymoma patients seem more susceptible to developing MG if their thymus displays GC in the adjacent thymic tissue [38].

## 2. Interferon Type I

### 2.1. IFN-I Expression

IFN type I (IFN-I) is composed of 13 IFN-α isoforms, and only one isoform for IFN-β, IFN-ω, IFN-ε, and IFN-κ, and all IFN-I genes are located on chromosome 9 in human [39]. IFN-I subtype genes share 30–85% homology within species [40]. All mammalian species express IFN-β and at least one IFN-α isoform [41]. 

In the case of pathogen infection, almost all the cells can produce IFN-I subtypes [42]. However, plasmacytoid DC are considered as the primary source of IFN-I and are also called “IFN-I producing cells” [43]. If plasmacytoid DC are the main source of IFN-I for systemic infections, they are not very effective in local infections [44]. Other cells can express IFN-I upon infection, such as macrophages, epithelial cells, neurons, or fibroblasts [45]. 

To produce IFN-I, cells must detect pathogen-associated molecular patterns (PAMP) associated with pathogen infection (bacteria or virus), such as part of bacterial wall proteins, lipids, or nucleic acids [46]. These PAMP are recognized by pattern recognition receptors composed of Toll-like receptors (TLR), RIG-like receptors (RLR), NOD-like receptors (NLR), and intracellular DNA sensors. Those receptors are localized at the cell surface, in endosomes or in the cytosol. Injured or dying cells can release endogenous molecules called damage-associated molecular patterns (DAMP) that can also trigger IFN-I production. DAMP can be nuclear components, such as endogenous DNA or RNA, or cytosolic proteins, and are recognized by the same pattern recognition receptors as PAMP [46,47]. In this case, it is called sterile inflammation because it is not caused by a pathogenic infection (Figure 1) [48].

IFN-I subtypes are expressed at a low level but have powerful activities. Their low level of expression makes them difficult to measure by standard experiments. However, a new and very sensitive method—the single-molecule array (SIMOA)—has enabled the measurement of blood concentration of IFN-α [49].

### 2.2. IFN-I Signalization

IFN-I can act in an autocrine, paracrine, or systemic way. All IFN-I subtypes bind a unique heterodimeric receptor: interferon-α/β receptor (IFNAR) 1 and 2 [50]. The length of interaction between IFN-I subtypes and their receptor is important for the diversity of biological activity responses. Antiviral activities are elicited after a brief IFNAR activation, while antiproliferative activities occur after a prolonged IFNAR activation [51]. 

IFNAR1 and IFNAR2 are associated with two cytoplasmic tyrosine kinases: tyrosine kinase 2 (TYK2) and Janus kinase 1 (JAK1), respectively. The first step of IFN-I signaling is the endocytosis of the IFNAR1-IFN-IFNAR2 ternary complex [52] that allows TYK2 and JAK1 to get closer and trans-phosphorylate each other. This leads to the recruitment of STAT1 and STAT2 by phosphorylation and STAT-activated proteins recruit IRF9 to form the ISGF3 complex, which migrates to the nucleus and binds the IFN-stimulated response element (ISRE) within the promoters of ISG (Figure 1) [40].

### 2.3. Retro-Control Mechanisms of IFN-I Signaling

To avoid chronic IFN-I production, IFN-I signaling is tightly controlled. An initial mechanism for reducing IFN-I signaling is the internalization, ubiquitination, and degradation of IFNAR1/2 after activation, which decrease the number of receptors on the cell surface [51]. 

Three major proteins are important in the feedback loop of IFN-I signaling: suppressors of cytokine signaling (SOCS) proteins SOCS1 and SOCS3, and the ubiquitin specific peptidase 18 (USP18). SOCS proteins play a role in the inhibition of JAK/STAT signaling whereas USP18 acts as a suppressor by binding IFNAR2 and competes with JAK1 [53]. Thus, it inhibits the IFN-I signaling cascade. USP18 seems to specifically inactivate IFN-α over IFN-β response [54].

IFN-I signalization can also be modulated by microRNA (miRNA). Forster et al. specifically reviewed miRNA implicated in IFN-I upregulation or downregulation signaling [55]. Briefly, miR-466 targets specifically IFN-α mRNA whereas miR-26a, miR-34a, and let-7b target *IFN-β* mRNA. miRNA can also act on *IFNAR1/2*. In mice, miR-29a reduces the expression of Ifnar1 on the surface of TEC. Loss of this miRNA induces the upregulation of Ifnar1, leading to hypersensitivity to PAMP and driving thymic involution [56]. Diverse miRNA can also modulate the expression of mRNA encoding for proteins involved in IFN-I signalization, such as *JAK1* [57], *STAT1/2* [58], and *SOCS1* [59].

All these different regulatory mechanisms are central to contain IFN-I signalization and avoid chronic inflammation.

### 2.4. Role of IFN-I in Innate and Adaptative Immunity 

Approximately 10% of the human genome is subject to IFN-I regulation [60]. IFN-I signaling induces the expression of more than 300 ISG that mediate the action of IFN-I subtypes [61,62]. de Veer et al. show the diversity of ISG and classify them into different categories such as cell-cell adhesion, host defense, nucleotide metabolism, antiviral, apoptosis, signaling, transcription factors, and immune modulation [62]. Some ISG are highly sensitive and induced by low amounts of IFN-I and others require high amounts of IFN-I. The first group is related to antiviral activities and the second to cell proliferation, chemokine activity, and inflammation [51]. Antiviral activities of ISG can act at all levels of viral infection: virus entry and trafficking, viral protein synthesis and genome amplification, and viral particle assembly and egress. 

The first description of the role of IFN-I in immunoregulation was their ability to regulate natural killer cell activities [63]. Since the impact of IFN-I on other immune cells has been largely demonstrated. For natural killer cells, Liang et al. have shown that IFN-α plays a potent role in mediating cell toxicity by upregulating perforin and Fas-L [64]. IFN-I subtypes play a key role in leukocyte adhesion and invasion to the infected site by inducing the expression of diverse chemokines [40]. In vitro, IFN-β induces CX3CL1 (Fractalkine) and CCL5 (RANTES) [65]. In vivo, IFN-I can favor the release of chemokines such as CCL2 and CXCL11, which recruit monocytes, memory T cells, and DC at the site of infection [66]. IFN-I can influence plasmacytoid DC by inducing antigen presentation and exacerbate their capacity for migration toward naive T cells. Some ISG are part of the proteasome and are responsible for antigen processing for the major histocompatibility complex class I [67]. IFN-I subtypes also have an impact on adaptive immune cells, i.e., T and B cells. In response to viral infection, CD8^+^ T cells require IFN-I signaling to generate effector and memory cells [68]. Additionally, IFN-I subtypes prevent the death of the activated T cells [69]. IFN-I can have contradictory effects on B cells: inhibiting the maturation and survival of immature B cells [70] but favoring B-cell activation, isotyping switching, and antibody production [71]. Altogether, these observations demonstrate that IFN-I molecules are not only anti-viral molecules, but they probably play a key role at the interface between innate and adaptative immunity as well. 

## 3. Implication of IFN-I in Diseases 

### 3.1. Type I Interferonopathies: Genetic Diseases

In 2011, Crow et al. defined a new group of diseases with the same features, i.e., autoinflammation and overexpression of IFN-I: the interferonopathies [72]. They are genetic diseases with childhood-onset, and 18 mutated genes have been identified in patients so far. The prototypic interferonopathy is the Aicardi–Goutières syndrome, a genetically heterogeneous disease with diverse gene mutations, all related to IFN-I signaling [73]. The seven potentially mutated genes associated with the Aicardi–Goutières syndrome are linked to either nucleic acid metabolism (*TREX1*, *RNAseH2A*, *RNAseH2B*, *RNAseH2C*, *SAMDH1*, or *ADAR*) or IFN-I signaling (*IFIH1*) [74]. Genes implicated in nucleic acid metabolism are mostly nucleases or editing enzymes (Figure 1). Seven other diseases compose the interferonopathy family: familial chilblain lupus, retinal vasculopathy with cerebral leukodystrophy, spondyloenchondrodysplasia, *STING* associated vasculopathy, *C1q* deficiency, Singleton Merten syndrome, and *USP18* deficiency [74].

All the genes associated with interferonopathies are involved in IFN-I production or signaling. Patients present an IFN-I signature in the periphery and also inflammation. IFN-α is a consistent marker of the Aicardi–Goutières syndrome, as it can be found in cerebrospinal fluid and sera [75]. Even if IFN-α tends to decrease during the course of the disease, ISG are still upregulated in peripheral mononuclear blood cells (PBMC) of patients [76]. IFN-α subtypes, and not IFN-β, seem to be the key feature of the disease because these patients develop only autoantibodies against IFN-α subtypes and not against IFN-β. However, no suitable test for the detection of IFN-β was available at the time of the disease characterization [77]. 

### 3.2. Acquired Interferonopathies: Autoimmune Diseases 

Some non-genetic autoimmune diseases are also considered interferonopathies as an IFN-I signature is observed. This is the case for several systemic autoimmune diseases. Systemic lupus erythematosus (SLE) is an autoimmune disease presenting multiple organ manifestations and characterized mainly by the presence of anti-nuclear antibodies. SLE was the first autoimmune disease suspected to be associated with IFN-I. An IFN-I signature is found in the blood of almost all children with SLE and 50% of adult patients [78]. Various findings have demonstrated the involvement of IFN-I in SLE: circulating immune complex can induce IFN-I production and polymorphism in genes involved IFN-I pathway has been identified in a GWAS [79].

Primary Sjögren’s syndrome is also characterized by an IFN-I signature in salivary glands and PBMC. In addition, ISG expression in PBMC correlates to titers of anti-Ro/SSA and anti-La/SSB autoantibodies, which are associated with primary Sjögren’s syndrome. Polymorphisms in genes linked to this IFN-I signature, in particular on IRF5, are associated with an increased risk for primary Sjögren’s syndrome [80].

An IFN-I signature is also detectable in PBMC of rheumatoid arthritis patients but to a lower level than that observed in SLE [81]. In addition, an increased expression of IFN-α, IFN-β, and ISG has been observed in the synovium of rheumatoid arthritis patients [82]. 

Patients with juvenile-onset dermatomyositis—an autoimmune disease characterized by muscle weakness, muscle inflammation, and skin rashes—present an overexpression of IFN-α in serum and an IFN-I signature in muscle and skin [49]. Moreover, the expression of ISG in muscle biopsies is correlated with the disease activity but the source of IFN-I remains unknown [83].

Altogether, this demonstrates that IFN-I subtypes are key cytokines in the pathogenesis of various systemic autoimmune diseases.

### 3.3. Iatrogenic effect of IFN-I

IFN-α or IFN-β are used as a treatment for various diseases and can induce transient autoimmune diseases. Several publications showed that IFN-I treatment can induce SLE [84]. The clinical onset can be from weeks to years. After the discontinuing of IFN-I treatment, patients stop displaying SLE symptoms [85]. IFN-α is used for chronic hepatitis C viral infection and some patients develop MG symptoms. Baik et al. published a history of all MG cases induced by IFN-α used as a treatment for chronic hepatitis C [86]. Those patients displayed different features of MG, such as autoantibodies against AChR, muscle weakness, and even respiratory crisis. In the case of MG-like symptoms, patients were given traditional treatment for MG [87]. IFN-α or IFN-β when given to patients with multiple sclerosis can also induce MG symptoms associated with ant-AChR antibodies [87,88,89].

The development of MG upon IFN-I treatment suggests that IFN-I could play a key role in the etiological mechanisms involved in MG.

## 4. Implication of IFN-I in Myasthenia Gravis 

### 4.1. No Obvious IFN-I Signature in the Periphery or the Muscle of MG Patients 

Chronic overexpression of IFN-I in the periphery is a hallmark of interferonopathies [72]. Using whole-transcriptome sequencing of PBMC from EOMG patients as compared to healthy controls, Barzago et al. show an increase in different inflammatory pathways related to ‘infectious disease’, ’inflammatory disease’, and ‘inflammatory response-associated’ but they did not look at IFN-I signature specifically [90]. Recently, our team searched for an IFN-I signature in the serum and PBMC of MG patients but no overexpression of IFN-I subtypes or ISG was detected in EOMG patients (Payet et al., unpublished data). In thymoma-associated-MG (MGT) no overexpression of IFN-I subtypes is detected in the periphery either but neutralizing autoantibodies against IFN-I subtypes are found [35,38]. 

Skeletal muscles are the target of AChR autoantibodies. The muscle is not a passive target, however no clear sign of inflammation has never been described in the muscle of MG patients. Increased expression of cytokines such as IL-6, IL-1, and TNF-α has been described but no IFN-I signature has never been observed either [91].

To conclude, no IFN-I seems to be detectable in the periphery or in the muscle of AChR-MG patients.

### 4.2. IFN-I in Early-Onset AChR-MG Patients

An IFN-I signature is detected in the thymus of EOMG AChR-MG patients and characterized by the overexpression of ISG [92] and an increased expression of IFN-β [93]. The ISG signature could also be due to IFN-II or IFN-III. An increased expression of IFN-II is detected in effector and regulatory T cells in MG patients [94]. As for IFN-III, *IL28A/B* mRNA expression is increased in MG thymuses [95]. Among these different IFN-I subtypes, IFN-β is suspected to orchestrate thymic changes in EOMG patients. Indeed, IFN-β induces in TEC the overexpression of α-AChR, the main antigenic target in MG, and this effect is much more powerful as compared to IFN-II and IFN-III [95]. Additionally, IFN-β slightly increases TEC apoptosis [95]. In co-culture experiments, pre-treatment of TEC with IFN-β increases the uptake of apoptotic TEC proteins by DC, which might lead to a cross-presentation of α-AChR by DC. This could explain the auto-sensitization against α-AChR and, therefore, autoantibody production. 

IFN-β also induced the overexpression of CXCL13 in TEC and of CCL21 in lymphatic vessels [25]. This could favor the recruitment of peripheral cells and GC development in the thymus of patients. Moreover, IFN-β is implicated in the overexpression of BAFF by TEC [95]. BAFF is a pro-survival cytokine for B cells, and its upregulation in the thymus could promote the survival of autoreactive B cells developing in the MG thymus.

The thymus of EOMG patients is inflammatory and characterized by the overexpression of diverse cytokines such as IL-1 [96], IL-6 [96], IL-17 [26], IL-23 [97], IFN-II [94], and TNF-α [94]. Among them, the overexpression of IL-23 has been recently demonstrated. This cytokine plays a major role in the differentiation of naïve CD4+ T cells toward pathogenic Th17 cells. IL-23 overexpression is associated with an increased number of Th17 cells in MG thymus and the production of Il-17. IFN-β that plays a key role in inducing IL-23 production by TEC could initiate a persistent thymic inflammation due to the reciprocal induction of IL-23 and IL-17 by TEC and Th17 cells, respectively. This may sustain the chronic thymic inflammation process and probably participate in autoantibody production [97]. 

Altogether, these observations demonstrate that IFN-β could be the orchestrator of thymic changes associated with early-onset AChR MG (Figure 2).

### 4.3. IFN-I in MG-Associated Thymoma

In MGT patients, autoantibodies against IFN-I subtypes are found in over half of patients, especially against IFN-α2, IFN-α8, and IFN-ω but are rarer for IFN-β [35]. Moreover, thymic extracted cells from the thymoma spontaneously express more anti-IFN-α2 antibodies than the rest of the non-thymomatous thymus [36], which indicates plasma cell infiltration in the thymoma. T-cell maturation process is impacted within thymoma and the production of autoantibodies could be related to AIRE deficiency in MGT [34]. 

The production of anti-IFN-I antibodies in MGT patients is not well understood. Nevertheless, this could be related to the specific overexpression of IFN-I subtypes by thymoma. This is observed only in thymoma from MGT patients and not in thymoma patients without MG or in the adjacent thymic tissue [98]. The overexpression of IFN-I subtypes might favor the autosensitization against these cytokines in MGT patients. The overexpression of IFN-α2 and IFN-β seems to be confined to the thymus because no overexpression of these cytokines is found in the sera of patients [38]. Altogether, these observations show a clear link between AChR MG and thymic IFN-I in MGT patients as in EOMG.

### 4.4. Impact of MG Treatments on the Thymic IFN-I Signature

As IFN-I seems to be the main orchestrator of AChR sensitization and thymic changes associated with MG, treatments that could reduce the effect of IFN-I would likely be of benefit.

Glucocorticoids can modulate IFN-I signaling pathways [99]. In MG, thymic overexpression of ISG is normalized in patients on glucocorticoids [100] and the decreased number of thymic GC could be directly linked to the inhibition of CXCL13 and CCL21 expression [95]. However, glucocorticoids do not efficiently inhibit all IFN-I effects. As an example, BAFF is still very high in IFN-I-treated TEC with or without glucocorticoid pretreatment, and BAFF level is not decreased in the thymus of MG patients on glucocorticoids [95]. Consequently, BAFF can still favor the development of autoreactive B-cells in these patients.

Thymectomy has to be considered the most efficient treatment to get rid of the IFN-I signature in AChR-MG. A randomized clinical trial has proven the beneficial effect of thymectomy on MG patients [101]. However, even if the site of inflammation is removed, autoreactive AChR- B cell clones that mature in the thymus and pass in the circulation can persist after thymectomy. These clones usually decline after thymectomy, but their persistence is associated with a worse disease status after thymectomy [102].

Immunotherapies could also affect IFN-I expression in the MG thymus in different ways. Rituximab, an anti-CD20 antibody appears to stimulate IFN-I production in B cells [103]. In contrast, eculizumab, a monoclonal antibody directed against complement component C5 might decrease IFN-I expression, as observed in patients with severe COVID-19 [104].

Altogether, this logically suggests that treatments that might downregulate IFN-I effects at the thymic level could be an additional beneficial therapeutic strategy.

### 4.5. Origins of IFN-I in MG Thymus

#### 4.5.1. Implication of Viral Infection in IFN-I Signature

Several mechanisms have been postulated to explain how pathogen infections might trigger autoimmunity: molecular mimicry or bystander effect [105]. The implication of viral infection in the triggering of MG is greatly debated [106]. As for other autoimmune diseases such as SLE, multiple sclerosis, and rheumatoid arthritis [107], several studies have investigated the potential link between MG and Epstein–Barr virus (EBV). This virus targets B cells and after the primo-infection, EBV establishes a life-long latency in memory B cells with occasional reactivation. The thymus is known to be a common target organ for viral infection [108]. Confirming results from an early study [109], Cavalcante et al. showed evidence of active EBV infection in MG thymuses with cells expressing EBV-encoded small RNAs (*EBER*) and EBV latency proteins [110]. Nevertheless, the observation of EBV-positive cells in MG thymus was not confirmed by two other independent investigations [111,112]. On the other hand, Csuka et al. showed that high levels of EBV nuclear antigen (EBNA)-1 antibodies are more common in MG than healthy controls and are particularly associated with EOMG [113].

The implication of EBV has also been investigated in thymoma. A systematic review of the literature about the role of EBV in the pathogenesis of thymomas suggests that EBV might only play a minor role [114]. Cavalcante et al. showed that some MGT patients are EBV-positive, but this seems to be linked to the MG and not specific to the thymoma. EBV positive thymus might be a consequence of EBV positive B-cell infiltration rather than causing thymoma [115].

It is not easy to establish a causal relationship between a worldwide infection such as EBV, and the triggering of a rare disease such as MG. In multiple sclerosis, it is suspected that autoimmunity would be rather associated with EBV infection in adulthood (infectious mononucleosis) rather than EBV infection in childhood [116,117]. In MG, it is also possible that EBV detection in the thymus is due to its reactivation in latent B cells in the inflammatory thymic environment and EBV could also locally sustain the thymic inflammation.

A link between MG and other viral agents has also been investigated. For example, poliovirus has been detected in thymic macrophages in some MG patients [118]. p16 (CDKN2A), a marker of human papillomavirus infections is upregulated in MGT. As papillomaviruses have a tropism for epithelial cells, they could be involved in thymoma but no sign of papillomaviruses was detected by RT-PCR [98]. To conclude, Leopardi et al. recently compile all the scientific publications referring to infectious agents. So far, there is no clear evidence that a specific pathogen infection causes MG [106].

#### 4.5.2. Activation of Innate Immune Signaling Pathways

To date, despite the IFN-I signature in the thymus of EOMG and MGT patients, the pathogenic contribution of viruses to MG has not been proven. However, the autoimmune disease onset can occur well after a possible triggering infection, when the pathogen might have already been cleared or the antiviral immune responses might have subsided (“hit-and-run” hypothesis) [119]. Nevertheless, the implication of innate immune signaling pathways has been clearly observed.

TLR play a critical role in pathogen recognition and activation of the innate immune response. Bernasconi et al. were the first to observe the increased expression of TLR4 in the thymus of EOMG patients [120]. TLR4 was first described to be activated by lipopolysaccharide (LPS) present in many Gram-negative and some Gram-positive bacteria. However, its ligands also include several viral proteins and various endogenous proteins [121]. In the MG thymus, TLR4 was selectively expressed on medullary TEC [120,122] and they suggest that TLR4 activation may induce the overexpression of IL-6, IL-12, and TNF-α and participate to thymic inflammation [120,122]. Later on, Cufi et al. observed the overexpression of TLR3 in EOMG thymuses, as well as the higher expression of protein kinase R (PKR) but not that of RIG-1 and melanoma differentiation-associated gene 5 (MDA5), proteins involved in double-stranded (ds)RNA recognition [93]. In thymoma from MG patients, an abnormal regulation of dsRNA-sensing molecules is also observed with an increase in TLR3 and a decrease in PKR, RIG-I, and MDA5 [98]. In addition, a specific overexpression for TLR6, TLR8, and TLR9 is also observed in hyperplastic MG thymuses, suggesting a link with the presence of GC [122].

To investigate whether TLR activation could trigger IFN-I expression in the thymus, TEC cultures have been treated with different TLR agonists. Cufi et al. demonstrated that Poly(I:C) the agonist of TLR3 greatly induces IFN-β that in turn triggers the selective expression of α-AChR in TEC and not that of other TSA [93]. Interestingly, mice treated with Poly(I:C) for one week present an overexpression of IFN-α2 and IFN-β in the thymus but also an increase in α-AChR. Moreover, IFNAR KO mice treated with Poly(I:C) do not overexpress α-AChR in the thymus, which means that overexpression of α-AChR is due to IFN-I signaling. The thymuses of mice treated with Poly(I:C) also display an increased expression of CXCL13, CCL21, and BAFF, like in human MG thymus [95]. After six weeks of Poly(I:C) injection, anti-AChR antibodies are found in the sera of mice but thymic changes are not observed anymore. Long-term treatments lead to the development of MG in mice characterized by muscular weakness and a decrease in α-AChR in diaphragm muscle [93]. 

Altogether, these observations suggest that the activation of TLR4 and TLR3/PKR in the thymus could be involved in the IFN-I signature and favor the development of MG as observed in the experimental MG mouse model [122]. The activation of these innate immune pathways is usually linked to pathogen infection but could also be due to endogenous molecules as TLR4 can be activated by low-density lipoproteins, beta-defensins, heat shock protein, and TLR3 by endogenous dsRNA [121]. This would explain why, to date, no pathogen infection has been clearly linked to MG onset.

#### 4.5.3. Altered miRNA Expression

miRNA are potent modulators of protein expression and are, therefore, involved in many physiological and pathophysiological processes. Specific miRNA are known to be involved in thymic pathogenesis associated with MG [123,124]. MIR-146, involved in retro-control of IFN-I signaling, is downregulated in the thymus of MG patients and this deficiency may contribute to persistent innate immune activation and inflammation [125]. By contrast, MIR-155 is upregulated in MG patients and could promote IFN-I signaling by targeting SOCS1 [126,127].

Papadopoulou et al. demonstrated that miRNA are essential in protecting thymic architecture. DICER is a key enzyme in miRNA biogenesis. Using conditional knock-out mice for Dicer in TEC (Foxn1Cre Dicerfl/fl), they observed a premature involution and the appearance of epithelial voids with dense foci of B cells. Furthermore, Dicer-deficient mice are hypersensitive to Poly(I:C) in line with the increased expression of Ifnar1 in TEC. This latter effect is mediated by the *miR-29a/b-1* cluster, as miR-29a targets *Ifnar1* and increases sensitivity to Poly(I:C) due to an increased expression of Ifnar1 in TEC [56]. These observations suggest a link between miR-29a and the IFN-I signature observed in the MG thymus. Cron et al. investigated the implication of DICER and the miR-29 family in thymic changes in EOMG. In the human thymus, decreased expression of DICER and MIR-29a-3p is observed in MG patients and also in TEC upon IFN-β treatment. However, it is not clear whether the decreased expression of MIR-29 subtypes in MG is either a consequence or a causative factor of the thymic inflammation. Nevertheless, the results indicate that a reduction in MIR-29 subtypes may contribute to the pathophysiological process involved in MG by favoring the emergence of pro-inflammatory Th17 cells [128].

#### 4.5.4. Potential Implication of Gene Polymorphisms in the Thymic IFN-I Signature in MG

Interferonopathies are associated with mutations in genes leading to chronic IFN-I expression. Could a genetic polymorphism affect the expression of a gene likely to play a role in the IFN-I signature detected in MG? Gene polymorphisms on *HLA* genes and other genes have been associated with increased risk for MG patients. Among them, the *PTPN22* (protein tyrosine phosphatase non-receptor type 22) gene is a susceptibility allele for some autoimmune diseases including MG [129]. PTPN22 positively regulates IFN-I expression upon TLR4 activation in myeloid cells. However, the variant *PTPN22W*—associated with MG—exhibits reduced function that cannot directly explain the IFN-I signature observed in MG patients [130]. Single nucleotide polymorphism in *TNIP1* (TNF-α-induced protein 3- (TNFAIP3-) interacting protein 1) has been associated with EOMG [131]. TNIP1 by binding to TNFAIP3 acts as a key repressor of inflammatory signaling, in particular downstream TLR activations [132]. Consequently, a *TNIP1* genetic variant could participate in a pathogenic IFN-I signature and the role of TNIP1 in MG should be further investigated.

## 5. Conclusions

Viral infection leads to an acute IFN-I signature to eliminate the virus. Interferonopathies are characterized by a chronic overexpression of IFN-I and autoinflammation [72]. No IFN-I signature seems to be detectable in the periphery or in the muscle of AChR-MG patients. As described above, the IFN-I signature is specifically located in the AChR-MG thymus.

This signature is found even long after the disease onset, suggesting a chronic IFN-I expression that could be due to a combination of factors involving a triggering event and an inadequate resolution of inflammation. The triggering event could be: (1) a pathogen infection but so far, no clear link with a pathogen infection could be established; (2) an event leading to an acute thymic involution leading to endogenous nucleic acids release and the activation of innate immunity signaling pathways; or (3) a combination of both events. In any case, the resulting induction of IFN-I expression is not effectively controlled and therefore leads to chronic IFN-I expression. As described in detail above, IFN-I seems to be the orchestrator of thymic changes favoring the development of AChR-MG. Even if the cause of this chronic IFN-I signature is not clear yet, this does not prevent considering MG as an organ-specific interferonopathy.

## Figures and Tables

**Figure 1 cells-11-01218-f001:**
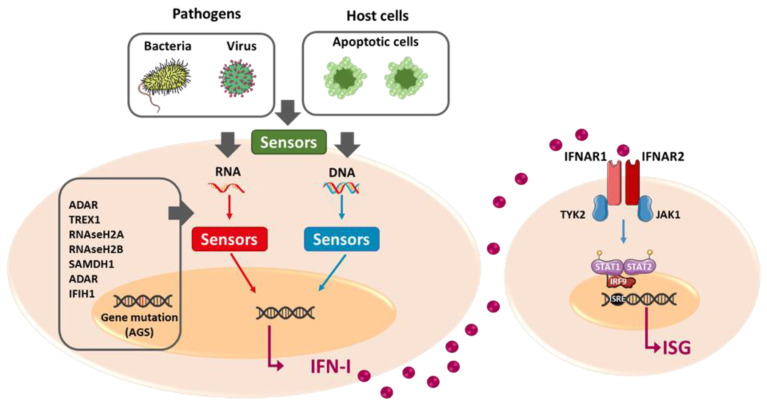
Mechanisms of IFN-I induction. IFN-I expression can be induced by diverse molecules from pathogens (bacteria or virus) or apoptotic host cells, PAMP and DAMP, respectively. These PAMP/DAMP are mostly nucleic acids (DNA and RNA) and are recognized by different cytosolic or membrane surface sensors. Their stimulation activates intracellular signaling cascades leading to the transcription of IFN-I subtypes. Once released, IFN-I subtypes interact with their receptor IFNAR1/2 that is ubiquitously expressed. Its stimulation induces the activation of TYK2 and JAK1, leading to the phosphorylation and heterodimerization of STAT1 and STAT2 that interact with interferon regulatory factor (IRF) 9. This complex, also known as IFN-stimulated gene factor 3 (ISGF3), then translocates into the nucleus and binds the IFN-I stimulated response element (ISRE) to regulate transcription of over 300 ISG. In interferonopathies such as AGS, genes involved in the metabolism of nucleic acids are mutated. This leads to the accumulation of endogenous nucleic acids and the abnormal stimulation of sensors triggering IFN-I expression and leading to the initiation and perpetuation of chronic inflammation.

**Figure 2 cells-11-01218-f002:**
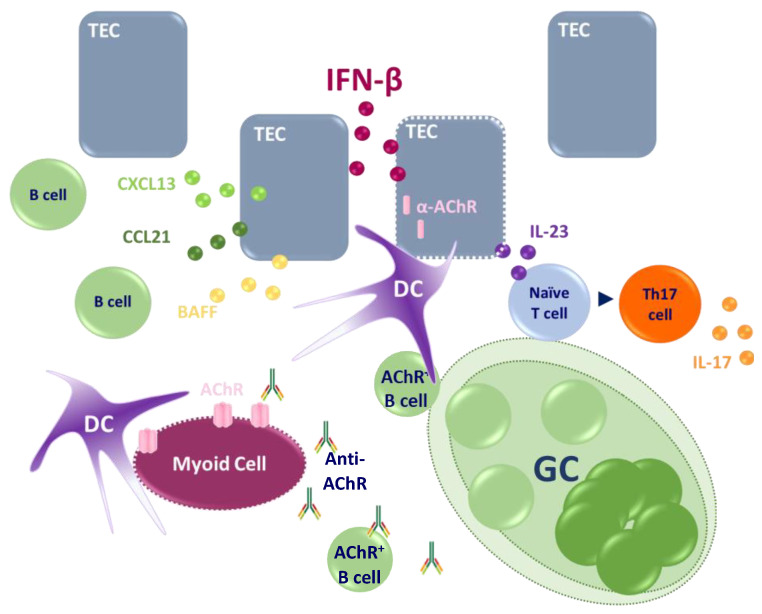
Implication of IFN-β in thymic changes associated with EOMG. IFN-β induces the expression of ⍺-AChR by TEC but tends also to induce TEC death. This could lead to the capture of TEC proteins, including the ⍺-AChR, by DC favoring an initial cross-presentation and autoreactivity toward ⍺-AChR. In parallel, IFN-β induces the expression of CXCL13 and CCL21, chemokines leading to the recruitment of peripheral B cells, and ectopic germinal center development. Germinal centers allow the maturation of autoreactive AChR B cells and their survival might be favored by BAFF that is overexpressed by TEC in response to IFN-β. Next, thymic myoid cells presenting AChR at their surface could be the first target of anti-AChR antibodies exacerbating the autoimmune reaction. IFN-β also induces pro-inflammatory cytokines such as IL-23 favoring the differentiation of naive T cells into pro-inflammatory Th17 cells that produce IL-17. Altogether, this demonstrates that IFN-β is a key orchestrator of thymic changes in EOMG.

## Data Availability

Not applicable.

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
