# Peer review of "Myasthenia Gravis: An Acquired Interferonopathy?"

_cells, 2022, doi:10.3390/cells11071218_

Round 1

Reviewer 1 Report

The paper entitled “Myasthenia Gravis: An Acquired Interferonopathy?” is well-written and provides a really interesting overview on interferonopathies and the role of IFN-I in myasthenia gravis.

I have only few comments:

-Paragraph 1.3: there is a repetition (“No alteration is observed in the thymus of 66 MuSK-MG patients [15] and the involvement of the thymus in LRP4-MG patients remains unclear, so far [9,16].  The thymus does not present tissue abnormality in MuSK-MG patients [15] and so far, the involvement of the thymus in LRP4-MG patients remains unclear [9,16]").

-The Authors should mention that MG associated with thymoma represents a more severe disease compared to non-thymomatous MG.

-The Authors should briefly discuss/speculate in a separate section on the possible effect of different available treatments for MG on IFN-I and on possible future strategies targeting IFN-I in specific MG subgroups.

".

Author Response

We thank the reviewer for its helpful comments. The manuscript has been improved based on the advice. Below in black are the reviewers' questions and our responses in bleu.

-Paragraph 1.3: there is a repetition (“No alteration is observed in the thymus of 66 MuSK-MG patients [15] and the involvement of the thymus in LRP4-MG patients remains unclear, so far [9,16].  The thymus does not present tissue abnormality in MuSK-MG patients [15] and so far, the involvement of the thymus in LRP4-MG patients remains unclear [9,16]").

The repetition has been removed at the end of paragraph 1.3 (page 4, lines 52-54)

-The Authors should mention that MG associated with thymoma represents a more severe disease compared to non-thymomatous MG.

These is now mentioned paragraph 1.3.2 (page 5, lines 96-97).

-The Authors should briefly discuss/speculate in a separate section on the possible effect of different available treatments for MG on IFN-I and on possible future strategies targeting IFN-I in specific MG subgroups.

As suggested by the reviewer, a new paragraph has been added between the initial ones 4.2 and 4.3, and the following staggered by as much (page 9, lines 258-270).

Reviewer 2 Report

This is a comprehensive and informative review of IFN-I expression, signalization, and its role in immunity.  Thymic function and existing evidence for INF-I's role in MG were also discussed.  I have the following comments and suggestions:

Line 26: References needed for statement regarding genetic predisposition and environmental risk factors.

Line 34: Reference needed for the classification of EOMG and LOMG

Line 37: ( ) needed around the phase "after the age of 65-70 years old" 

Line 41: AchR abs are found in ~85% of generalized MG patients.  "in EOMG and LOMG patients" at the end of that sentence is redundant. 

Lines 49-50:  Rivner et al (Muscle Nerve 2020;62:333-343) found that 70% of patients with LRP4 antibodies were MGFA class III (moderately severe), IV (severe), V (in crisis), so they are not mildly affected. 

Lines 115-118:  need editing to clarify what this statement meant to convey

Lines118-119:  need rewording.  Thymoma is characterized by a deficit in AIRE relative to the surrounding thymic tissue?

Line 121: Is "affecting peripheral tolerance" at the end of the sentence needed? (Since the next sentence explains the effect)

Line 130: homology "within" or across species?

Figure 1: Arrow missing to link PAMP/DAMP to membrane surface sensors.  ISRE mentioned in legend but not labeled in figure.

Lines 163-166:  Instead of describing what happens in interferonopathies, which is not the topic of this review, perhaps the authors might want to explain/explore how ISG are related to the pathogenesis of MG?

Lines 232-282: Since this review's focus is on the role of IFN-I in MG, the information on genetic inferferonopathies and other autoimmune diseases, while interesting, can be substantially abbreviated to keep the spotlight on MG.

Lines 349-357: My understanding is that the circulating anti-AchR antibodies are produced by plasma cells outside the thymus.  Since IFN-I is not detected peripherally or at the NMJ, a discussion of how IFN-I related changes in the thymus/thymoma lead to peripheral production of anti-AchR antibodies would close the loop for the readers.  

Lines 396-399: Incomplete sentence

Lines 401-412:  It is hard to follow how these reported findings fit into the proposition that TLR play a critical role in pathogen recognition and activation of the innate immune response.  A more detailed explanation linking these observations into a coherent hypothesis rather than presenting them as a collection of findings would be tremendously helpful.

Lines 453-488 (conclusion): The conclusion section needs extensive revision.  The authors introduced genetic mutations and polymorphisms as an explanation for the observations described in previous sections.  This information should be presented as a section by itself.  In my opinion, the authors should conclude with a synthesis the available data to explain the role of IFN-I in the pathogenesis of MG.  Identify what crucial data are missing to support the role of IFN-I in the pathogenesis of MG and what the "take home" points are.   

Wolfe et al (N Engl J Med 2016; 375:511-522) reported thymectomy had positive effects on the course of MG.  Please comment on how this intervention impacts the IFN-I pathway in MG.

Author Response

We thank the reviewer for the helpful comments. The manuscript has been improved based on the advice. Below in black are the reviewers' questions and our responses in bleu.

Line 26: References needed for statement regarding genetic predisposition and environmental risk factors.

A reference has been added in paragraph 1.1 (page 4, line 17):  Avidan et al., J. Autoimmunity 2014 (doi:10.1016/j.jaut.2013.12.001).

Line 34: Reference needed for the classification of EOMG and LOMG

A reference has been added in paragraph 1.1 (page 4, line 25):  Cortés-Vicente et al., Neurology 2020 (doi:10.1212/WNL.0000000000008903).

Line 37: ( ) needed around the phase "after the age of 65-70 years old" 

Parentheses have been added to the sentence (page 4, line 27).

Line 41: AchR abs are found in ~85% of generalized MG patients.  "in EOMG and LOMG patients" at the end of that sentence is redundant. 

The sentence has been modified (page 4, line 31).

Lines 49-50:  Rivner et al (Muscle Nerve 2020;62:333-343) found that 70% of patients with LRP4 antibodies were MGFA class III (moderately severe), IV (severe), V (in crisis), so they are not mildly affected. 

Rivner et al. analyzed anti-AChR and MuSK negative patients (DNMG) that were anti-LPR4 positive patients. In the 24 patients identified, 23 were also anti-agrin positive (LAPMG). This group of LAPMG patients was compared to the DNMG group and not to groups of AChR or MuSK patients. It is consequently difficult to compare the severity of these LAPMG patients to AChR or MuSK MG patients or even “pure” LRP4 MG patients. It appears that we do not have enough knowledge on LRP4 MG patients to clearly conclude about the severity and the term “mild” has been removed to avoid any misinterpretation (page 4, line 38).

Lines 115-118:  need editing to clarify what this statement meant to convey

Lines118-119:  need rewording.  Thymoma is characterized by a deficit in AIRE relative to the surrounding thymic tissue?

Line 121: Is "affecting peripheral tolerance" at the end of the sentence needed? (Since the next sentence explains the effect)

With respect to these last three reviewer comments, the paragraph was carefully checked and rewritten (pages 5-6, lines 99-107).

Line 130: homology "within" or across species?

Homology between IFN subtypes is within species (page 6, line 112).

Figure 1: Arrow missing to link PAMP/DAMP to membrane surface sensors.  ISRE mentioned in legend but not labeled in figure.

The figure has been modified (page 6).

Lines 163-166:  Instead of describing what happens in interferonopathies, which is not the topic of this review, perhaps the authors might want to explain/explore how ISG are related to the pathogenesis of MG?

Lines 232-282: Since this review's focus is on the role of IFN-I in MG, the information on genetic inferferonopathies and other autoimmune diseases, while interesting, can be substantially abbreviated to keep the spotlight on MG.

With respect to these two comments of the reviewer, we believe that information on interferonopathies is essential to understand the hypothesis behind this review. In addition, the role IFN-I in the pathogenesis of MG is detailed in paragraph 4.1 and a review of the effect of potentially 300 ISG might be too much for this review.

Lines 349-357: My understanding is that the circulating anti-AchR antibodies are produced by plasma cells outside the thymus.  Since IFN-I is not detected peripherally or at the NMJ, a discussion of how IFN-I related changes in the thymus/thymoma lead to peripheral production of anti-AchR antibodies would close the loop for the readers.  

Taking into account the reviewer 1's comment, we wrote a new paragraph (4.4, page 11, lines 333-353). In this paragraph, we have developed the idea suggested by the reviewer 2. In particular the fact that even if there is apparently no IFN-I signature in the periphery, thymic inflammation favors the development of autoreactive AChR B cell clones that can persist in the periphery.

 Lines 396-399: Incomplete sentence

The line numbers are different from the original version, but we think that the reviewer was mentioning the second sentence of the original paragraph 4.5.2. The word « since » has been removed (page 12, line 387).

Lines 401-412:  It is hard to follow how these reported findings fit into the proposition that TLR plays a critical role in pathogen recognition and activation of the innate immune response.  A more detailed explanation linking these observations into a coherent hypothesis rather than presenting them as a collection of findings would be tremendously helpful.

The paragraph 4.5.2 has been modified to take into account the advice of the reviewer (page 12, paragraph 4.5.2)

Lines 453-488 (conclusion): The conclusion section needs extensive revision.  The authors introduced genetic mutations and polymorphisms as an explanation for the observations described in previous sections.  This information should be presented as a section by itself.  In my opinion, the authors should conclude with a synthesis the available data to explain the role of IFN-I in the pathogenesis of MG.  Identify what crucial data are missing to support the role of IFN-I in the pathogenesis of MG and what the "take home" points are.   

We fully agree with the reviewer and the section on genetic polymorphisms has been moved above to a new paragraph (paragraph 4.5.4, page 13). Another paragraph has been created to deal with the absence of IFN-I signature in the periphery and in the muscle of MG patients (new paragraph 4.1, page 9). The conclusion is now more focused on our original hypothesis: Is Myasthenia Gravis an acquired interferonopathy?

Wolfe et al (N Engl J Med 2016; 375:511-522) reported thymectomy had positive effects on the course of MG.  Please comment on how this intervention impacts the IFN-I pathway in MG.

This has been commented in paragraph 4.4 (page 11, line 344-345)

Reviewer 3 Report

The manuscript entitled "Myasthenia Gravis: An Acquired Interferonopathy?" aimed to review the pathogenic role of IFN-I considering the thymus alterations in ACHR-MG. This review manuscript is for the special issue of the "Cells" dedicated to the memory of Dr. Pia Bernasconi who was a well-known scientist for her studies about thymus TLRs  and their contribution to the pathogenesis of MG. 

Therefore, the manuscript mentioned MG,  ACHR-MG association with thymus, IFN-I, the implication of IFN-I in diseases and the relation of IFN-I with ACHR-MG. The authors discussed the points they intended to make, intelligibly. They reviewed the literature that is up-to-date and relevant to the manuscript.The figures were adequate and straightforward.

The authors have been contributing to the research field of MG and autoimmunity with highly-qualified studies for a long time. In this manuscript, they analysed clearly whether the ACHR-MG could be an acquired interferonopathy with pointing out significant studies. They mentioned their very recent study (unpublished data)  showed that no overexpression of IFN-I subtypes or ISG was detected in EOMG patients. Another studies they mentioned were confirming that no overexpression of IFN-I subtypes is detected in the periphery in MGT either but neutralizing autoantibodies against IFN-I subtypes are found. So, overall they concluded the manuscript that the cause of  chronic IFN-I signature is not clear yet but this does not prevent from considering MG as an organ-specific interferonopathy. This conclusion is very coherent and reasonable with the literature they referred.

I would like to note  that on page 2, lines between 66-70; the adjacent sentences seem to repeat themselves in meaning. Choosing one of the sentences may make the context more understandable.

This review manuscript may be beneficial to researchers that are interested in MG, autoimmunity, the relation between ACHR-MG and thymus and interferonopathies.

The manuscript can be accepted. 

Author Response

We thank the reviewer for the helpful comments. The manuscript has been improved based on the advice. Below in black are the reviewers' questions and our responses in bleu.

I would like to note  that on page 2, lines between 66-70; the adjacent sentences seem to repeat themselves in meaning. Choosing one of the sentences may make the context more understandable.

The repetition has been removed at the end of paragraph 1.3 (page 4)

Round 2

Reviewer 2 Report

The current manuscript shows that the authors had addressed issues raised in the initial review.  I would like to suggest the following additional edits:

Line 38: insert "with" after "present" to read "present with ocular or generalized MG,..."

Delete lines 55 and 56 (redundant)

Line 59: Thymic changes are clearly observed in EOMG patients but not in LOMG without thymoma, probably as because the thymus strongly significantly involutes with aging.

Line 94: "90% of thymoma patients develop an autoimmune disease, 30% of which are AChR-MG patients with MG associated with AChR-antibodies (AChR+MG)

Lines 97-99:   "B1 or B2 thymoma are characterized by an increased density of immature thymocytes, few medullar compartments especially for B2 thymoma subtype which probably leads to an impaired negative selection of thymocytes." Need clarification: previously the authors mentioned that thymus cortex is associated with positive selection.  This statement seems at odd with that previous statement.  

Author Response

We thank the reviewer for the helpful comments. The manuscript has been improved based on the advice. Below in black are the reviewers' questions and our responses in blue.

The current manuscript shows that the authors had addressed issues raised in the initial review.  I would like to suggest the following additional edits:

Line 38: insert "with" after "present" to read "present with ocular or generalized MG,..."

The sentence has modified line 38

Delete lines 55 and 56 (redundant)

We do not see the redundancy because the sentence refers to MuSK MG patients, and then LRP4 MG patients.

Line 59: Thymic changes are clearly observed in EOMG patients but not in LOMG without thymoma, probably as because the thymus strongly significantly involutes with aging.

The sentence has modified line 59

Line 94: "90% of thymoma patients develop an autoimmune disease, 30% of which are AChR-MG patients with MG associated with AChR-antibodies (AChR+MG)

The sentence has modified line 94

Lines 97-99:   "B1 or B2 thymoma are characterized by an increased density of immature thymocytes, few medullar compartments especially for B2 thymoma subtype which probably leads to an impaired negative selection of thymocytes." Need clarification: previously the authors mentioned that thymus cortex is associated with positive selection.  This statement seems at odd with that previous statement.  

In the normal thymus, positive and negative selections occur in thymic cortical and medullary regions, respectively. In B1 and B2 thymomas, the presence of immature thymocytes undergoing positive selection is observed. Single positive CD4 or CD8 T differentiate. However, the emergence of autoreactive T cells is probably due to altered negative selection process afterward in a thymic disordered microenvironment.

The sentence has been modified to be clearer lines 97-101